



# Mid-level clouds are frequent above the southeast Atlantic stratocumulus clouds

**Adeyemi A. Adebiyi[1], Paquita Zuidema[2], Ian Chang[3], and Sharon P Burton[4]**

[1]Department of Atmospheric and Oceanic Sciences, University of California Los Angeles.

[2]Rosenstiel School of Marine and Atmospheric Sciences, University of Miami, FL.

[3]School of Meteorology, University of Oklahoma, Norman, OK

[4]NASA Langley Research Center, Hampton, Virginia 23681, USA

*Correspondence to*: Adeyemi A. Adebiyi (aadebiy@ucla.edu)

**Abstract.**

Shortwave-absorbing aerosols seasonally overlay extensive low-level stratocumulus clouds over the southeast Atlantic. While a lot of attention has been focused on the interactions between the low-level clouds and the overlying aerosols, no study has yet focused on the mid-level clouds that also occur over the region. The presence of mid-level clouds over the region complicates the attribution of the cloud radiation budget, as well as of space-based remote-sensing retrievals. Here we characterize the mid-level clouds over the southeast Atlantic using lidar- and radar-based satellite cloud

retrievals in addition to the observations collected in September 2016 during the ORACLES (ObseRvations of Aerosols above CLouds and their intEractionS) field campaign. We find that the mid-level clouds over the southeast Atlantic are relatively common, with the overwhelming majority of the cloud occurring between altitudes of 5 and 7 km and temperatures of 0 and -20 ℃. These clouds occur at the top of a moist mid-tropospheric smoke aerosol layer, most frequently between August and October, closer to the southern African coast than farther offshore, and more frequently

during the night than during the day. Between July and October, we find that about 64% of the mid-level clouds have a geometric cloud thickness less than 1 km, and about 60 % have a cloud optical depth less than 4. Using the lidar-based depolarization-backscatter relationship for September 2016, we find that the mid-level clouds are liquid-only clouds with no evidence of the existence of ice. Furthermore, we also find that these clouds are mostly associated with synoptically-modulated mid-tropospheric moisture outflow that can be linked to the detrainment from the continental-

based clouds. Overall, the presence of these supercooled mid-level clouds influences the regional cloud radiative budget by reducing the radiative cooling rates by about 10 K/day near the top of the more-dominant low-level clouds.





## 1 Introduction

Clouds over the southeast Atlantic are important to reducing the uncertainties in the global climate because they account
for one of the world's five major subtropical stratocumulus cloud regions (Klein and Hartmann, 1993). Alone, these
stratocumulus clouds cool the global climate system because they predominantly reflect the incoming shortwave
radiation, and exert a small effect on the outgoing longwave radiation (Wood, 2012). In addition, the stratocumulus
clouds over the southeast Atlantic are different from others, because they are accompanied by the presence of elevated
smoke aerosol layers in September and October, when free-tropospheric zonal winds emanating off of continental
Africa are a maximum (Adebiyi and Zuidema, 2016). This aerosol circulation pattern can strengthen the underlying
low cloud deck through either meteorological or aerosol influences (Johnson et al., 2004; Wilcox, 2010; Adebiyi and
Zuidema, 2018; Gordon et al., 2018; Deaconu et al., 2019). This interaction between the elevated smoke aerosols and
stratocumulus clouds has received substantial attention in recent years from the research community, because of its
unique impact on both regional and the global climate (Boucher et al., 2013). While the aerosol-stratocumulus-cloud
interactions over southeast Atlantic complicate the estimation of the cloud radiative forcing, a recent study also
highlights the presence of high moisture content that accompanies the smoke transport above the southeast Atlantic
low-level clouds (Adebiyi et al., 2015). The occurrence of this high mid-tropospheric moisture points to the likelihood
of mid-level clouds over the southeast Atlantic, which has not been highlighted in previous literature. The recent
ORACLES (ObseRvations of Aerosols above CLouds and their intEractionS) field campaign (Redemann et al., 2020)
observed such mid-level clouds (Figure 1). Their location within and at the top of the smoke layer suggests potential
interaction with the smoke aerosols  (e.g. Lohmann and Feichter, 2005) that could further complicate the estimation of
the cloud radiative forcing over the region. Whereas stratocumulus clouds tend to cool the regional climate,
unaccounted mid-level clouds with colder cloud tops and warming cloud radiative effects may likely offset the cooling
effect associated with the stratocumulus clouds (Christensen et al., 2013; Bourgeois et al., 2016). Therefore, an accurate
picture of the multi-layer cloud system occurring in the presence of an elevated smoke layer is necessary to fully
understand the complexity of the radiative interactions over the region.

Sub-tropical mid-level clouds have received little to no attention when compared to those over equatorial or mid-
latitude regions (Bourgeois et al., 2018, 2016; Fleishauer et al., 2002; Riihimaki et al., 2012; Riley and Mapes, 2009;
Stein et al., 2011). Global distribution of mid-level clouds covers about 25% of the Earth's surface (Sassen and Wang,
2012) and account for about 30% of all clouds (Zhang et al., 2010). With significant land-ocean contrast, the cross-
section of the mid-level cloud fraction generally increases from the tropical oceans to the mid-latitude regions
(Bourgeois et al., 2016; Zhang et al., 2010, 2005). In contrast to the mid-latitude region, there is a higher occurrence
of nighttime mid-level clouds than during daytime over the tropics (Zhang et al., 2010). Regardless of their location in
the tropics or the mid-latitude, mid-level clouds mostly consist of supercooled liquid water, with most studies placing
the temperature at the top of the clouds between ~0ºC and -15ºC, thickness typically less than 2 km, and cloud-top
heights between 4-8 km (Bourgeois et al., 2018; Riihimaki et al., 2012; Riley and Mapes, 2009; Stein et al., 2011).
With ice particles likely forming at a temperature less than -6ºC (Hobbs et al., 1985), the 0 – to – -15ºC temperature





range of the optically-thin mid-level clouds suggests mixed-phase microphysics are possible (Zhang et al., 2010), with potential impact in both shortwave and longwave spectrum.

Despite its significance, climate models have found it difficult to simulate the distribution and properties of these mid-level clouds accurately, and observational constraints by passive satellite sensors can be biased in multi-layer cloud regions. Specifically, models consistently underestimate the mid-level clouds by simulating less than 40 % of the observed global distribution (Zhang et al., 2005). One reason for this underestimation is the misrepresentation of potential mixed-phase processes, whereby liquid-water droplets and ice crystals may coexist and persist for long
periods of time, thus presenting a unique challenge for global model parameterizations (e.g. Liu and Krueger, 1998). Another reason for the underestimation of mid-level clouds is because most models find it difficult to simulate multi-type multi-layer cloud systems (Tselioudis and Kollias, 2007), thus overestimating high clouds due to their lack of detrainment of moisture by convection schemes at the mid-troposphere (Bodas-Salcedo et al., 2008). While cloud retrievals from space-based passive satellite sensors are often used as validation and opportunity to improve these
models, they also suffer in regions with multi-layer cloud scenes (Holz et al., 2009). For passive sensor aboard satellites like the Meteosat-10 or MODIS (Moderate Resolution Imaging Spectroradiometer), multi-layer cloud scenes often provide top-of-atmosphere radiances that are either too cold for the lower-level clouds, or too warm for the upper-level clouds retrievals (Davis et al., 2009), thus introducing uncertainties in the retrieved cloud properties.

In contrast, active remote-sensing measurements such as those from lidar and radar instruments are useful to easily
identify the mid-level clouds and their properties in multi-layer cloud scenes (Figure 2a). These active sensors can be part of a ground-based station, mounted on an aircraft or a space-borne satellite. Over the southeast Atlantic, lidar measurements of clouds and aerosol vertical distributions were made on aircraft during the first phase of the NASA ORACLES field campaign in September 2016 (Redemann et al., 2020). These measurements provided the first airborne observation of the mid-level cloud and confirmed its prevalence over the southeast Atlantic. However, these
measurements only covered a short period and made it difficult to characterize the climatological state of the clouds over the region. Lidar and radar instruments aboard the CALIPSO (Cloud-Aerosol Lidar and Infrared Pathfinder Satellite Observations; (Winker et al., 2003)) and CloudSat (Stephens et al., 2002) satellites respectively provide continuous spatial coverage and useful retrievals of clouds and aerosols over the southeast Atlantic. The combined information from CALIPSO and CloudSat provides a unique dataset that gives a reliable detection of the multi-layer
cloud system and its properties over the southeast Atlantic (Mace and Zhang, 2014). In this study, we use the ORACLES aircraft measurements, CALIPSO-only, and the CloudSat-CALIPSO merged datasets to document the characteristics and properties of the mid-level clouds above the southeast Atlantic stratocumulus clouds.

## 2. Data and Methods

We define the mid-level clouds as clouds above the low-level clouds and between 3 km and 8 km over the southeast
Atlantic. These altitude levels correspond to the standard pressure of approximately 700 to 350 hPa. Our definition of the mid-level cloud is consistent with previous studies (e.g., Riihimaki et al., 2012; Bourgeois et al., 2016, 2018), and we use it here also because the inversion-capped low-level clouds are generally topped below ~3km over the southeast



Atlantic (e.g. Painemal et al., 2014; Adebiyi et al., 2015). We focus our analysis primarily on the region between 10ºW-10ºE and 5º-20ºS, which is approximately the region that was covered by the 2016 ORACLES field campaign, but it

is also the region dominated by climatological low-level clouds between July and October over the southeast Atlantic (Zuidema et al., 2016). Despite the focus on this delimited region, our analysis broadly considers the occurrence of mid-level clouds over the entire southeast Atlantic.

We primarily use the cloud information measured by the second generation airborne High Spectral Resolution Lidar (HSRL-2) aboard the NASA ER-2 aircraft during the September 2016 ORACLES field campaign (hereafter called

ORACLES-2016). ORACLES-2016 was conducted out of Walvis Bay in Namibia. Unlike other ORACLES subsequent deployments in August 2017 and October 2018 that operated out of São Tomé and Príncipe, only ORACLES-2016 deployed the ER-2 aircraft, capable of reaching 70,000 feet (21.3 km) in altitude and hosting the HSRL-2 lidar that year (Redemann et al., 2020). HSRL-2 measures backscatter, extinction and depolarization ratio of atmospheric constituents at 355 and 532 nm and also the backscattering and depolarization ratio at 1064 nm (Burton

et al., 2018). The vertically-resolved multi-wavelength and depolarization measurements of mid-level clouds are invaluable for accurately distinguishing the altitude and phase of the mid-level clouds, providing a unique view of the multi-layer cloud and aerosol system over the southeast Atlantic. Details of the instrument, calibrations, and algorithms can be found in Burton et al. (2015, 2018) and the references therein. Of the twelve ER-2 flight-days conducted during ORACLES-2016, each between 7–9 h in duration, HSRL-2 was active for seven days. We use the HSRL-2 version R7

data with a vertical resolution of 15 m and a horizontal resolution of 10 seconds or approximately 1.8 km. Therein, we primarily use the HSRL-2 cloud-top heights, aerosol extinction, particulate backscatter, and particulate depolarization ratio information at 532 nm. We also use the temperature information included in the dataset and from the Modern-Era Retrospective analysis for Research and Applications, Version 2 (MERRA-2; Gelaro et al., 2017) reanalysis that is collocated to HSRL-2 measurements. A further ancillary dataset are the *in-situ* measurements gathered by the P-3 plane

within and below the mid-level clouds depicted in Figure 1.

A regional climatology of the mid-level cloud properties over the southeast Atlantic was developed from the cloud retrievals from the CloudSat and CALIPSO products (Stephens et al., 2002; Winker et al., 2003). Both CloudSat and CALIPSO are part of the A-train constellation, with footprints overlapping by more than 90 % of the time (Stephens et al., 2008). While CloudSat carries a 94 GHz Cloud Profiling Radar (CPR), CALIPSO carries the Cloud-Aerosol

Lidar with Orthogonal Polarization (CALIOP). Although the instruments are built differently, both are able to observe the atmospheric vertical distributions, with the CALIOP lidar more sensitive to the aerosols and optically-thin clouds than the CPR radar (Mace and Zhang, 2014). In contrast, the CPR radar is suitable for an optically-thick cloud layer, and it is able to determine the phase and other microphysical properties of the cloud better than the CALIOP lidar (Sassen and Wang, 2008). The combined product thus provides unique data useful to understand the macro- and micro-

physical characteristics of the mid-level clouds above the optically-thick stratocumulus clouds.

In this study, we use the CALIOP-only retrievals to determine the height level of the mid-level clouds, including the cloud-top heights, and we use the CloudSat-CALIPSO merged dataset to analyze the essential cloud properties. We obtain the mid-level cloud-top heights and aerosol extinction at 532 nm wavelength from version-3 of level-2 CALIOP





*Layer_Top_Altitude* and *Extinction_Coefficient_532* products. Although there were some improvements in the version
4 cloud–aerosol discrimination algorithm, most of them were specifically focused on aerosol lofted into the upper
atmosphere or the lower stratosphere (Liu et al., 2019). As a result, more than 95% of all aerosol and cloud layers
detected within the troposphere remain largely unchanged (Liu et al., 2019) between versions 3 and 4. Using the
*Layer_Top_Altitude* product, we determine the cloud-top height (km) as the mid-level cloud layer top, and the the
frequency of occurrence of mid-level cloud as the number of CALIOP profiles with observed mid-level clouds to the
total number of observations over the regions. Furthermore, we also obtained three products from the merged CloudSat-
CALIPSO datasets: *2B-GEOPROF-LIDAR* which provides the fraction of hydrometeor in each layer (Mace and Zhang,
2014), *2B-TAU* which provides the cloud optical depth, and *2B-FLXHR-LIDAR* which provides the estimates of
broadband fluxes and radiative heating rates in the atmospheric column (L'Ecuyer et al., 2008; Henderson et al., 2013).
Specifically, we obtained the *LayerTop* and *LayerBase* variables from *2B-GEOPROF-LIDAR* product to determine the
heights, frequency of occurrence, and the geometric thickness (top minus base) of the mid-level cloud layers defined
between 3 km and 8 km; the *layer_optical_depth_2B_TAU* variable from the *2B-TAU* product to determine the cloud
optical depth which was averaged for the identified mid-level cloud layer; and the QR_2B_FLXHR_LIDAR from the
*2B-FLXHR-LIDAR*  product to determine the heating rates at the top of the low-level clouds. For the latter, the
underlying low-level clouds are defined for cloud layers identified below ~3 km. In addition, the low-level cloud-top
heating rates are assessed for cases when there are collocated overlying mid-level clouds and when there are none.

While the level-2 CALIOP products are reported at a horizontal resolution of 5 km and vertical resolution between 60
and 360 m (Hunt et al., 2009), the combined products have a horizontal resolution of approximately  1.3 km by 1.7 km,
and the effective vertical resolution at nadir is 240 m (Stephens et al., 2008). Although CALIOP retrieval extends up
to the present day, we analyze the mid-level cloud properties only between 2006 and 2010, where both sensors measure
the atmospheric volume within 15 seconds from each other and high-quality products are available for both the
CALIOP-only and CloudSat-CALIPSO merged products. We ignore data after 2010 because of battery anomaly that
caused the CloudSat satellite to stop collecting data and eventually lost formation with the A-train constellation in 2011
(Nayak et al., 2012). While CloudSat rejoined A-train in June 2012, it is positioned in a different satellite constellation
such that its observing time is 100 seconds different from CALIOP. Furthermore,  CloudSat only acquired
measurements during the daytime in the post-anomaly period, resulting in about a 50% reduction in the sampling size
compared to the pre-anomaly period (Mace and Zhang, 2014).

In addition to the cloud information from ORACLES and CloudSat-CALIPSO merged datasets, other datasets helped
characterize the variability of the mid-level clouds and their large-scale environment. We obtain the temperature,
moisture, and wind information over the southeast Atlantic region from ECMWF ERA-Interim reanalysis (Dee et al.,
2011), in addition to ECMWF auxiliary data that is interpolated to CloudSat-CALIPO bins (Partain, 2007). We also
obtain additional cloud and aerosol information of daily-averaged retrievals from the Moderate Resolution Imaging
Spectroradiometer (MODIS) and the Meteosat-10 Second Generation (MET10) satellites. Specifically, we obtained
the MODIS-*Terra* low-level cloud fraction and aerosol optical depth retrievals (King et al., 2013) as well as cloud-top
heights and brightness temperature from the Spinning Enhanced Visible and Infrared Imager (SEVIRI) instrument
onboard MET10 satellite (Schmetz et al., 2002). In particular, we use the cloud information from SEVIRI to assess the





diurnal and spatial variability of the mid-level clouds. Although a passive instrument such as SEVIRI has difficulty accurately capturing the cloud top heights when a mid-level cloud is above an optically-thick low-level cloud (Figure 3; see also fig. S-1), they are useful because of their broad swaths and high temporal resolution. SEVIRI aboard the geostationary MET10 satellite sits at 35,786 km altitude centered at approximately 9.5° E longitude, with cloud

observation at a temporal resolution of 15 min and 3 km spatial resolution at the sub-satellite point. We use both the brightness temperature directly retrieved at 10.8 μm infra-red channel and the cloud-top heights retrieved using the NASA-Langley cloud product algorithm. While details can be found in Minnis et al. (1995), this algorithm combines techniques that use the information which spans from visible (0.65 μm) to infra-red (10.8 μm) channels to obtain improved retrieval accuracies (Palikonda et al., 2006).

**3. Results**

An example of the vertical profile of the total attenuated backscatter from CALIPSO shows that the southeast Atlantic features not only the presence of the elevated smoke and the low-level clouds, but also the mid-level clouds (Figure 2a). For this example, these mid-level clouds are between 12S-18S, and they significantly attenuate the lidar signal directly below them (see more CALIPSO images in supplementary Fig. S-2). Therefore, we document here the

occurrences (section 3.1), the properties (section 3.2), the associated large-scale meteorology (section 3.3), the radiative impacts on the low-level clouds (section 3.4), and the diurnal variability (section 3.5) of the mid-level clouds to provide a full picture of the complicated cloud-aerosol system over the southeast Atlantic.

### 3.1. Occurrence of mid-level clouds over the southeast Atlantic

During the ORACLES-2016 campaign, mid-level clouds were observed in 5 out of 7 days that the HSRL-2 was active

over the southeast Atlantic (Figure 2b). Although the observation of mid-level clouds occurs over most parts of the campaign region, their cloud-top heights are generally below 7 km (see inset in Figure 2b). To better understand what the preferred altitude levels are for the mid-level clouds, we estimate the probability distributions of the HSRL-2 cloud-top heights and compare that with those from CALIOP during September over the campaign region (Figure 2b). We found that the majority of the HSRL-2 mid-level cloud top heights occur between 5-7 km, with the median value at

approximately 5.4 km and the probability distribution collectively reaching up to about 25 %. Although CALIPSO overpasses in September 2016 do not directly correspond to the locations and time of the HSRL-2-inferred mid-level clouds, they similarly show that the clouds appear to have a preferred altitude between the 5-7 km range. In fact, the CALIOP distribution of the mid-level cloud climatology for all September between 2006-2010 agrees remarkably well with the distribution that uses only the values in September 2016 or for the few days of the HSRL-2 observations.

Overall, about 93% of the mid-level cloud-top heights measured during ORACLES-2016 are above 5km, compared to ~77% and ~61% from the CALIOP-derived mid-level clouds respectively for September 2016 and September 2006-2010.

The mid-level clouds typically occur in the presence of smoke aerosols over the southeast Atlantic. As the CALIOP attenuated backscatter example shows in Figure 2a, the smoke aerosols are typically found immediately below the mid-

level clouds, although there are cases where the clouds form inside the elevated smoke aerosol layer (see also Figure 1





and fig. S-2). Indeed, further analysis of HSRL-2 and CALIOP extinction profiles in September shows that the averaged aerosol extinction coefficients over a 1-km layer immediately below the mid-level clouds are about 0.21 and 0.14 $km^{-1}$, respectively (Figure 2c). While the effects of humidity on aerosols scattering is likely possible (e.g. Magi and Hobbs, 2003), these values are markedly higher than the 0.03 and 0.10 $km^{-1}$ for the corresponding 1-km layer above the clouds

for HSRL-2 and CALIOP respectively. The HSRL-2 profiles in September 2016 indicate a cleaner layer above the mid-level clouds, with aerosol extinction coefficient decreasing to zero faster than those observed from CALIOP (Figure 2c). As the CALIOP image in Figure 2a also suggests, the mid-level clouds and the smoke layer also occur over a region that is usually covered by underlying warm-liquid clouds during September (e.g., Adebiyi et al., 2015). Although beyond the scope of this study, the presence of the mid-level clouds and the smoke aerosols over the region

presents a unique and exciting new challenge for aerosol-cloud interaction studies.

While the ORACLE-2016 and CALIPSO observations shown in Figure 2 are for September, the mid-level clouds over the southeast are also present in other months. Figure 4a shows that the mid-level clouds are more prevalent between August and October compared to other months. The frequency of occurrence – estimated hereafter as the number of profiles the mid-level clouds are observed to the total number of observations – shows a minimum of about 2 % in

June and a maximum of about 15 % in September when averaged over the ORACLES-2016 campaign region (5º20ºS and 10ºW-10ºE; inset in Figure 2b). Furthermore, the seasonal cycle of the mid-level clouds overlaps with that of the smoke aerosol loading, further highlighting the co-occurrence of the smoke aerosols below the mid-level clouds. Of particular interest is the time period between July and October because that is when the smoke aerosol loading, and the underlying low-level cloud fraction also reach their climatological maximum (Figure 4a). Therefore, we examine the

spatial distribution of the mid-level cloud regional climatology between July and October (Figure 4b & c). We find that the mid-level clouds are common near the coast, with a frequency of occurrence of up to 30 % that gradually decreases westward (Figure 4b). For those above the climatologically high low-level cloud region north of 20ºS (black contour in Figure 4b), the mean mid-level cloud-top heights are overwhelmingly between 5 and 6 km. In contrast to the north of ~20ºS, the mid-level clouds south of 20ºS occur less frequently by about 10-15 %, with the mean cloud-

top heights higher by about 1 km (Figure 4b & c). This contrast between the mid-level clouds north and south of ~20ºS highlights the complexity and variability of the cloud systems over the southeast Atlantic. Unlike north of 20ºS, which is dominated by separated low-level and mid-level clouds, the cloud system south of 20ºS often occurs as a unified deep-convective cloud system extending from low- to upper-level atmosphere as part of the eastward-traveling mid-latitude disturbances (Adebiyi and Zuidema, 2018). As a result, the isolated mid-level cloud is not as common south

of 20ºS as it is north of 20ºS.

### 3.2. Properties of the mid-level clouds

We focus on the region north of 20ºS and examine the properties of the mid-level clouds using the observations during the ORACLES-2016 campaign and the merged CloudSat-CALIPSO datasets. We analyze the probability and the cumulative distribution (Figure 5a-c) of the mid-level cloud optical depth, its geometric thickness (km), and cloud

temperature (ºC). Similar to the CALIPSO-only analysis, the majority of the mid-level cloud-top heights for the merged CloudSat-CALIPSO datasets between July and October is also between 5-7 km (compare Figure 2 & fig. S-3a).



Furthermore, the result shows that the geometric thickness and the cloud optical depth (a measure of extinction of solar radiation by clouds) are predominantly less than 1 km and 4, respectively (Figure 5a & b). Specifically, about 64% of the mid-level clouds have a cloud thickness that is less than 1 km (85% for a thickness of less than 1.5 km), and about 60 % have a cloud optical depth that is less than 4 (72 % for an optical depth of 6). These results suggest that the mid-level clouds over southeast Atlantic are geometrically and optically-thin clouds. For comparison, the same thickness in stratocumulus clouds could have a cloud optical depth greater than 20 (e.g. Szczodrak et al., 2001; Haywood et al., 2004).

In addition to the southeast Atlantic mid-level clouds being optically-thin, these mid-level clouds also have distributions that span warm to cold temperatures. Figure 5c & 4d show the temperature distribution of the mid-level clouds, respectively obtained for the CloudSat-CALIPSO merged dataset between July and October (2006-2010) and for the mid-level clouds observed during ORACLES-2016. For both cases, the temperature distributions generally extend between -20 °C to about 4 °C, with the majority of the mid-level clouds colder than 0 °C. Specifically, about 98 % and 87 % of the mid-level clouds obtained from the field campaign and merged CloudSat-CALIPSO datasets respectively are below the 0 °C temperature (grey lines in Figure 5c & d). In addition, the majority of the cold mid-level clouds are observed above 5 km, evident in the CloudSat-CALIPSO (Figure 5c), and in the HSRL-2 datasets with observed mid-level clouds generally above 5 km (Figure 5d and Figure 2b). Furthermore, the mid-level clouds also show double peaks in the probability distribution for both CloudSat-CALIPSO and HSRL-2 datasets: one around -4 °C and the other around -9 °C. For the CloudSat-CALIPSO dataset, the warmer peak (~ -4 °C) corresponds to mid-level cloud-top heights less than 6 km, while the colder peak (~ -9 °C) corresponds to cloud-top heights higher than 6 km (Figure 5c). This double peak in temperature distribution over the southeast Atlantic is similar to those documented for tropical and mid-latitude mid-level clouds (e.g. Riihimaki et al., 2012; Riley and Mapes, 2009). However, one notable difference is that our second temperature peak (~ -9 °C) is markedly warmer than previously reported for other regions (which is typically between -12.5 °C and -20 °C). Potential reasons for this difference are explored in section 3.3.

Regardless of the temperature, mid-level clouds over the southeast Atlantic are dominated by supercooled liquid water. Figure 6 shows the relationship between the 532 nm lidar depolarization ratio and backscatter for mid-level clouds obtained from HSRL2 and CALIOP in September 2016 (blue dots). The depolarization-backscatter relationship has previously been used for cloud phase discrimination (Hu et al., 2007, 2009), because of its distinct relationship for ice (high-level clouds; cloud-top heights greater than 8 km; green dots in Figure 6) and liquid-water clouds (low-level clouds; cloud-top heights less than 3 km; red dots in Figure 6). Unlike the high-level clouds, the relationship between the depolarization and backscatter for low-level cloud is largely positively correlated, since they are predominantly spherical liquid-water clouds. We find that the mid-level clouds observed during the ORACLES-2016 (Figure 6a) follow the depolarization-backscatter signature of a low-level cloud, indicating that the mid-level clouds are liquid-water only with no presence of ice. Although these observations are obtained at the mid-level cloud tops, the layer-mean observations from CALIOP also shows similar depolarization-backscatter relationship (Figure 6b). Despite the difficulty of lidar-based instruments in detecting very low concentrations of the ice crystals (e.g. Bühl et al., 2013) in a high liquid-water environment, the few observations of the mid-level clouds that fall within the depolarization-



backscatter relationship of high-level clouds are likely due to the uncertainties in the off-nadir measurements by
CALIOP (e.g. Hu et al., 2009). Overall, the mid-level clouds over the southeast Atlantic are optically-thin, super-cooled
liquid-water clouds.

The one in-situ characterization of a mid-level cloud and its aerosol environment just below the cloud base occurred
on 4 September 2016 (Figure 1 and fig. S-4). The cloud top temperature was -4C, with a cloud top temperature inversion
of 5 degrees providing a strong stability cap. The icing of most of the cloud probes ultimately did not allow for much
data acquisition on the cloud microphysical properties, but what exists suggests the droplets were likely extremely
small, discouraging their ability to glaciate. The effective radius measured by the Two-dimensional Stereo (2D-S)
cloud probe did not exceed that of its smallest size bin, 5 μm, indicating a lack of precipitation (consistent with visual
notes taken during ORACLES; Redemann et al., 2020). High aerosol concentrations will also support a Twomey effect.
The Passive Cavity Aerosol Spectrometer Probe (PCASP), measuring all particles with diameters between 0.1 to 3.0
μm, indicates cloud condensation nuclei concentrations of approximately 700 $cm^{-3}$ m below the cloud. Condensation
nuclei concentrations for particles with diameters greater than 3 (10 nm) were approximately 3500 (3000) $cm^{-3}$ below
the cloud. Although black carbon is hydrophilic, the organic aerosol that contributes most of the biomass-burning
aerosol mass (sub-cloud organic aerosol mass concentrations were approximately 20 micrograms $m^{-3}$) support the
ability of the biomass burning aerosol to serve as a cloud nucleator (Kacarab et al., 2020). As indicated in Figure 1,
sufficient turbulence is released within the cloud to occasionally support its development above the capping inversion.

### 3.3. Large-scale meteorology associated with the mid-level clouds.

Over the southeast Atlantic region, both large-scale subsidence and the presence of shortwave-absorbing smoke
aerosols will warm the free troposphere during the July-October period, and it is evident that any presence of mid-level
cloud must be supported by a large-scale environment that is conducive for its development. Here, we examine this
large-scale meteorology and the associated mechanism that may help to sustain the mid-level clouds after its formation
over the southeast Atlantic. Unlike the semi-permanent low-level clouds that consistently receive a steady supply of
moisture in the boundary layer from the underlying ocean, the mid-level clouds lack a consistent moisture source, and
are more susceptible to variations in environmental conditions. As such, observational evidence from either CALIPSO
or CloudSat satellites may not be sufficient to capture the dynamical impacts of the large-scale environment because
of the poor spatial coverage (e.g. CloudSat footprint is ~1.4 km by 2.5 km) and temporal resolution (16 days return
period). In contrast, geostationary satellites, such as the Meteosat-10 satellite, provide broader coverage of the southeast
Atlantic with higher temporal resolution (~15 minutes). Therefore, they are better at providing insights into the
dynamical evolution of the mid-level clouds. One major problem with passive sensors on geostationary satellites,
however, is that they generally suffer in multi-layer cloud regions, such that cloud-top height retrievals are lower than
observed by lidar-based satellites (e.g. Hamann et al., 2014). For example, the cloud-top height retrieval from SEVIRI
onboard Meteosat-10 satellite on 22nd September 2016 is about 1-2 km lower than that from the nearby CALIPSO
overpass (compare Figure 2a and Figure 3). However, it also captures other occurrences of the mid-level clouds over
the southeast Atlantic that are not within the CALIPSO footprint. Despite the inaccuracies in identifying the mid-level
cloud-top heights, we nonetheless use the observations from SEVIRI here because of their broader coverage and higher



temporal resolution to better understand the impacts of the large-scale meteorology on the evolution of the mid-level clouds.

Therefore, we explore the possible mechanisms that support the occurrence of mid-level clouds north of 20 ºS, by first considering the coupling of the offshore mid-level clouds with the adjacent southern African continent. Unlike the south of 20ºS, where the mid-level clouds are associated with the mid-latitude westerly disturbance that supports the
southern hemisphere storm tracks (e.g., Hoskins et al., 2005), the large-scale dynamical regime associated with the mid-level clouds north of 20ºS is expected to be different (e.g. Adebiyi and Zuidema, 2018). Figure 7a shows an example Hovmöller diagram of mid-level clouds identified by the SEVIRI's brightness temperature for for 11-20 September 2016 and overlaid with moisture flux (black contour) and easterly zonal wind speed (grey contour) calculated using ERA-Interim reanalysis values averaged between 3-8 km. The figure shows that while the clouds
associated with the convective system over land are common, there are occasional offshore mid-level clouds over the ocean that are accompanied by strong moisture flux that propagates westward (see also fig. S-5). For this example, two major moisture outflow events occur – one between approximately 11-16$^{th}$ and the other after 18$^{th}$ of September 2016. In both cases, the moisture fluxes reaching more than 30 g m kg$^{-1}$ s$^{-1}$ are accompanied by zonal winds reaching more than 6 m s$^{-1}$. This anecdotal evidence is useful to understand the large-scale progression that highlights the connection
between the offshore mid-level clouds and the continental moisture outflow.

Between July and October, the climatology of this mid-tropospheric moisture flux further indicates that the southeast Atlantic mid-level clouds are associated with the deep-layer moisture of the convective regime over the Congo-Zaire basin (Figure 7b). The moisture flux vectors north of ~20S overwhelmingly point to the moisture transporting in the westward direction (Figure 7b), coincident with the climatology of strong mid-level winds previously identified in
Adebiyi and Zuidema, (2016). As a result, the spatial region of maximum mid-level moisture flux corresponds to the maximum region of the southern African easterly jet (compare Figure 7b to fig. 4 in Adebiyi and Zuidema, 2016). Furthermore, we estimate the moisture flux divergence over land north of 20S (blue shade in Figure 7c) and found that it can be associated with the moisture convergence occurring directly offshore (red shade in Figure 7c), where the mid-level clouds occur most frequently between July and October (compare Figure 7c with Figure 4b). This suggests that
the offshore mid-level clouds are likely either detrained from the convective system over land or they are generated at the top of a continental boundary layer previously moistened by convection, before advecting offshore under the influence of the strong zonal winds.

Even in the absence of advected mid-level clouds, the advection of moisture not yet reaching a relative humidity of 100% can also generate an isolated mid-level cloud through radiative cooling. High relative humidity within the mid-
troposphere can result in increased longwave cooling for the upper part of the layer and contemporaneous warming in the lower part of the layer (e.g. Larson et al., 2006). This differential heating can set-off a process that results in turbulent mixing, which can redistribute moisture to the upper part of the layer, and in turn, strengthen radiative cooling, thus leading to the development of mid-level clouds. Figure 7d shows the vertical distribution of the offshore moisture flux convergence. Strong convergence of moisture and the potential for strong turbulence and instability directly below
0ºC level can promote the development of mid-level clouds with tops between the 0ºC and the -20ºC isotherm (cf.





Figure 5c & d). Furthermore, it is noteworthy that a smoke layer almost always co-occurs with the mid-level clouds observed during ORACLES-2016, (see Figure 1, Figure 2, and fig. S-2), suggesting that presence of smoke particles likely contributes to the presence of the mid-level cloud over the southeast Atlantic. One way the presence of the smoke can aid the development of the mid-level cloud is by strengthening the turbulent mixing within the layer through, for

example, a preferential warming in the lower part of the layer (e.g. Adebiyi et al., 2015). As a result, the co-occurrence of the moisture and smoke aerosols within the layer serves as an ideal recipe to generate an isolated mid-level cloud characterized by strong mixing within the layer and strong radiative cooling at the top. Whether the presence of an advected mid-level cloud and the associated cloud-top longwave cooling can foster turbulent mixing below the cloud, or it is the turbulent mixing associated with the moisture and shortwave-absorbing smoke aerosol layers that result in

the development of the mid-level cloud is beyond the scope of this study. In the same way, it is beyond the scope of this study to determine which share of the observed mid-level clouds can be attributed to one process or the other. Nevertheless, it is clear that the presence of a high-humidity environment and the associated effect of longwave radiative cooling likely contribute to the development and the eventual sustainability of the mid-level clouds over the southeast Atlantic.

**3.4. Radiative impact of the mid-level clouds on the low-level clouds**

Because the low-level clouds dominate the southeast Atlantic between July and October, it is useful to examine the radiative impact of the mid-level clouds on the underlying low-level clouds during the same period. Figure 8 shows the low-level cloud-top instantaneous heating rates obtained from the merged CloudSat-CALIPSO dataset between July and October (2006-2010) when the mid-level cloud is present above the low-level clouds, and when they are not.

Details of how the heating rates are estimated for the CloudSat-CALIPSO datasets can be found in Henderson et al. (2013, and references therein). The low-level clouds are defined here as a cloud layer with tops less than 3 km. Because of the cloud-top temperature and the high liquid water content, there is typically a stronger longwave cooling than there is shortwave heating near the tops of the low-level clouds. Over the southeast Atlantic between July and October, this results in the mean shortwave radiative heating rates of 5 K/day and longwave cooling rate of -21 K/day, for a net

cooling rate of ~ -16 K/day (Figure 8a).

The presence of mid-level clouds over the southeast Atlantic, however, reduces the net radiative cooling substantially at the top of these low-level clouds. In the shortwave, this reduction is due primarily to the decrease in the downwelling radiation reaching the low-level cloud top as a result of the mid-level cloud. Consequently, this leads to an overall reduction in the shortwave heating rate near the top of the low-level cloud by about 2 K/day. For longwave, the presence

of the mid-level clouds increases the downwelling radiation that reaches the top of the low-level cloud, thus leading to a reduction in the longwave cooling rate by about 12.5 K/day. Thus, the presence of mid-level clouds reduces the net cooling rate near the top of the low-level cloud by about 10.5 K/day, which is approximately a 65 % reduction in the net radiative cooling rates (Figure 8a).

There is potentially a chance that the mid-level clouds lead to overall warming at the top of the low-level cloud. That

is because the downwelling longwave flux reaching the top of the low-level cloud is largely proportional to the mid-



level cloud optical depth. Thus, increases in the mid-level cloud optical depth result in increases in the downwelling longwave fluxes and in decreases in the net radiative cooling rates at the top of the low-level cloud (Figure 8b). For sufficiently high mid-level cloud optical depth (~ 11), the shortwave heating surpasses the longwave cooling, resulting in net radiative heating rates, rather than cooling, at the top of the low-level clouds. This is mitigated by the contrasting

circulation patterns for the two cloud levels, and further work is required to indicate if a lasting effect is present on the underlying cloud development.

### 3.5. Diurnal variations of the mid-level clouds

The impacts of longwave radiative cooling, while always present, are more obvious at night when shortwave warming is not occurring. In addition, the indication that the offshore mid-level clouds are associated with moisture detrainment

from convection over land, which has a separate distinctive diurnal cycle (Bourgeois et al., 2016), motivates an examination of the diurnal variability of the offshore mid-level cloud and its relationship to that of clouds over land. Figure 9 shows the frequency of occurrence of the mid-level clouds averaged over the ocean (0–10ºE) and over the land (10ºE–20ºE) obtained from CALIOP and SEVIRI. Over both the ocean and land, more mid-level clouds are observed during the nighttime than daytime. This result is consistent for both CALIOP and SEVIRI, although the

frequency of occurrence is significantly lower in the case of SEVIRI because of the difficulty of observing the mid-level clouds (e.g., Figure 3). Nevertheless, as in the case of Figure 7a and because of the fine 15-min temporal resolution of the mid-level clouds, we use SEVIRI here only to capture the structure of diurnal variability and not its magnitude. The magnitude of the frequency of occurrence from CALIPSO shows about ~8 % (~28 %) during daytime (nighttime) over land, and ~7 % (~12 %) over the ocean. When accessed at the approximate overpass time of CALIPSO, which is

between 12:30-13:30 UTC during the day and 00:30-1:30 UTC during the night, the ratio of the daytime occurrence to the nighttime occurrence from SEVIRI is about 39 % over the ocean, which is about 17% lower than obtained from CALIPSO. SEVIRI further indicates that the mid-level clouds maximize between 03-05 UTC in the morning and minimize between 11-13 UTC in the afternoon (Figure 9b). Furthermore, despite the difference in the frequency of occurrence between daytime and nighttime, the probability distribution of the mid-level cloud-top heights obtained

from CALIPSO are largely similar (see supplementary fig. S-6).

Overall, the diurnal variability in the amplitude of mid-level cloud occurrence is modulated by the competing influence of the longwave and shortwave radiative heating. The cloud-top longwave cooling and the associated instability are expected to dominate during the night, while the shortwave heating, subsidence, and cloud dissipation are expected to compete with the longwave cooling, and possibly dominate, during the day. The weaker diurnal cycle over the ocean

coupled with a lower occurrence of the mid-level cloud is consistent with less free tropospheric moisture over the ocean than over land, and it affirms that the continent is the source of the offshore moisture.

### 4    Discussions and Conclusions

The southeast Atlantic is an important region because it features one of the major subtropical stratocumulus clouds below one of the most extensive elevated smoke-aerosol layers in the world. While a lot of attention has been focused

on the low-level stratocumulus clouds due to their abundance, persistence, and regional climate impacts, as well as the





aerosol-cloud interactions that are associated with the elevated smoke aerosols and the stratocumulus clouds, no study has yet characterized the mid-level clouds that also occur over the southeast Atlantic. The presence of mid-level cloud over this region could complicate the attribution of regional cloud radiative effects, and the region's contribution to the global radiative budget. Previous studies have mostly focused on the characteristics of mid-level clouds over the

equatorial and mid-latitude regions, with little attention given to mid-level clouds over the sub-tropical region. Here we document the characteristics of the mid-level clouds over the southeast Atlantic stratocumulus cloud region, using a combination of aircraft and satellite observations, as well as reanalysis datasets.

Our analysis primarily relies on the observations of the mid-level cloud collected during September 2016 of the NASA ORACLES field campaign from the HSRL-2 aboard the NASA ER-2 aircraft. Unlike other ORACLES subsequent

deployment that utilizes P-3 aircraft (August 2017 and October 2018), ER-2 aircraft used in September 2016 was capable of reaching about 70,000 feet (21.3 km) in altitude, with instruments onboard able to observe the entire vertical column of the atmosphere and thus provided a unique view of the multi-layer cloud over the southeast Atlantic. This dataset was extended with satellite observations that include the retrievals of cloud properties from CALIPSO-only and CloudSat-CALIPSO merged datasets between 2006 and 2010 as well as cloud observations from the Meteosat-10

Second Generation (MET10) geostationary satellites and environmental variables from ECMWF reanalysis dataset.

Our result shows that the mid-level clouds over the southeast Atlantic are relatively common, with the cloud-top heights typically placed between 5 and 7 km. Measurements from the HSRL-2 indicate that about 93% of the mid-level clouds observed during the ORACLES campaign are above 5km. Between 2006 and 2010, the CALIOP-derived mid-level clouds indicate that the majority (about 61 %) of the mid-level cloud-top heights are similarly found between 5-7 km

altitude, suggesting a preferred altitude layer for the mid-level clouds. In addition, the monthly-averaged CALIOP frequency of occurrence indicates that the mid-level clouds are prevalent between August and October, with the maximum occurring in September (about 15 % of the time) and the minimum in June (about 2 % of the time). The results further indicate that the frequency of occurrence over the southeast Atlantic is highest near the coastal region up to about 30 %, with a gradual decrease westward when averaged between July and October (2006-2010). This period

of maximum occurrence of the mid-level clouds also corresponds to the period when the elevated smoke aerosol loading and the low-level cloud fraction maximizes over the southeast Atlantic. This co-occurrence thus highlights the significance of mid-level clouds in influencing the radiative impacts both within the smoke layer and on the underlying low-level clouds over the region. Furthermore, our analysis shows that the aerosol extinctions immediately below the mid-level cloud are markedly higher than those above it, showing that the mid-level clouds tend to mostly occur at the

top of the moist, smoke-aerosol layer.

Between July and October, our results indicate that the mid-level clouds over the southeast Atlantic are optically-thin and are characterized by supercooled liquid-water clouds. Specifically, about 64% of the mid-level clouds have a cloud thickness that is less than 1 km (about 85% for a thickness of less than 1.5 km), and about 60 % have a cloud optical depth that is less than 4 (72 % for an optical depth of 6). In addition, the probability distribution of the temperature of

the mid-level clouds shows that they occur predominantly between 0 ºC and -20ºC. Indeed, the temperature distribution collocated with HSRL2-observed mid-level clouds during the September-2016 ORACLES campaign indicates that





more than 98 % of the clouds have temperature between 0 °C and -20°C, which is also comparable with the percentage of mid-level clouds below 0°C (87 %) that are collocated with the merged CloudSat-CALIPSO datasets between July and October (2006-10). Despite the cold temperature range, mid-level clouds observed by HSRL-2 and CALIOP-only instruments during September 2016 places the 532 nm depolarization-backscatter relationships within the signature expected for liquid-water clouds, suggesting no presence of ice in the mid-level clouds over the southeast Atlantic.

Furthermore, we find that the mid-level clouds over the southeast Atlantic are mostly associated with synoptically-modulated continental moisture outflow, which can be linked to the detrainment from the continental convective clouds. Using ERA-Interim reanalysis, our analysis shows strong moisture convergence offshore that can be associated with deep-layer moisture of convective regime over the Congo-Zaire basin, and a strong mid-tropospheric zonal wind associated with the southern African easterly jet (Adebiyi and Zuidema, 2016) over the southeast Atlantic. In addition, we also highlighted the possibility of the mid-tropospheric high-humidity layer, in the presence of smoke aerosols, over the southeast Atlantic generating an isolated mid-level cloud due to potential turbulent mixing within the layer and strong radiative cooling at the top of the layer. The impacts of radiative cooling, while always present, are more obvious at night when shortwave warming is not occurring. Indeed, the merged CloudSat-CALIPSO dataset shows that the mid-level cloud frequency of occurrence averages about 12 % during nighttime over the ocean (5°S-20°S, 0-10°E), compared to only about 7 % during the daytime. The overall diurnal variability over the ocean is consistent with those over land, with the maximum occurring between 03-05 UTC in the morning and minimum occurring between 11-13 UTC in the afternoon.

The presence of these mid-level clouds impacts the radiation reaching the top of the underlying low-level clouds. Between July and October, our analysis shows the presence of the mid-level clouds results in about 2 K/day reduction in the shortwave heating rates and about 12.5 K/day reduction in the longwave cooling rates near the top of the underlying low-level clouds. This reduction in heating rates is mainly due to the reduction of the downwelling longwave radiation when the mid-level clouds are present. Overall, there is about a 10.5 K/day reduction in the net radiative cooling rates associated with the presence of the mid-level clouds, which accounts for about a 65 % reduction when compared to the case without overlying mid-level clouds. The radiative impact of mid-level clouds on the underlying low-level clouds depends on many factors, including the mid-level cloud-top heights, the cloud-base heights, cloud optical depth, temperature, and the microphysical compositions of the mid-level clouds. It also depends on the concentration of smoke aerosols that is between the mid-level and low-level clouds. For example, we examined the dependence of the low cloud-top radiative cooling rates on the mid-level cloud optical depth. The results indicate that the low cloud-top radiative cooling rates decrease almost proportionally with increases in the mid-level cloud optical depth. Beyond a mid-level cloud optical depth of ~11, the shortwave heating rates surpass the longwave cooling rates, leading to a net radiative heating rate, rather than cooling, near the top of the low-level clouds. Thus, the implication of reduced net radiative cooling, or the net radiative warming, near the top of the low-level clouds, suggests



that the presence of mid-level clouds will likely facilitate a decrease in turbulent mixing within the boundary layer. This must be weighted by the amount of time the mid-level cloud is present over a particular low cloud scene.

Overall, the radiation reaching the surface or, more significantly, the top of the atmosphere will be impacted by the presence of the mid-level clouds, despite the presence of the elevated smoke and the low-level clouds. Furthermore, while our analysis highlighted the higher aerosol extinction coefficients below the mid-level clouds than above it, it
does not however examine the potential influence of the smoke-induced shortwave warming on the development, dissipation, or lifetime of the mid-level clouds over the region, since it is beyond the scope of this study. The presence of elevated smoke aerosol below (or around) the mid-level clouds strongly points to the potential for aerosol-cloud interaction in the cold environment. Despite the substantial radiative impacts of the smoke aerosols and low-level clouds, our main conclusion here is that the mid-level clouds over the southeast Atlantic are non-negligible, and any
uncertainty in the representation of their properties will likely contribute to the uncertainties in the regional cloud radiative effects.

**Data availability**

All HSRL-2 data can be found at the ORACLES ESPO archives
(https://espoarchive.nasa.gov/archive/browse/oracles/id8/ER2). We obtain the CALIOP data from the Atmospheric Science Data Center (ASDC) assessible at https://eosweb.larc.nasa.gov/project/calipso/calipso_table. The merged CloudSat-CALIPSO data, including the ECMWF auxiliary dataset, can be obtained directly from CloudSat Data Processing Center (http://www.cloudsat.cira.colostate.edu/). The SEVIRI datasets are obtained from the Satellite ClOud and Radiation Property retrieval System (SatCORPS) as part of the NASA Langley support for 2016
ORACLES campaign (https://satcorps.larc.nasa.gov/cgi-bin/site/showdoc?docid=4&cmd=field-experiment-homepage&exp=ARM-ORACLES).

**Author contribution.**

A.A.A designed the project. A.A.A performed the analysis with contributions from P.Z. and I.C. A.A.A wrote the
paper. All authors discussed the results and commented on the manuscript.

**Competing interests.**

The authors declare that they have no conflict of interest.

**Acknowledgements.**

This work was developed with support from the University of California President's Postdoctoral Fellowship awarded to A.A.A., and the NASA Earth Venture Suborbital-2 ORACLES grant NNX15AF98G awarded to P.Z. We thank Leonhard Pfister for helpful comments and discussion. We also thank the HSRL team for making the HSRL-2 datasets available for use in this paper. We also thank Patrick Minnis for providing access to the SEVIRI datasets as
part of the NASA Langley for the ORACLES campaign. Finally, we thank leadership and the participants of the ORACLES 2016 campaign.





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

705



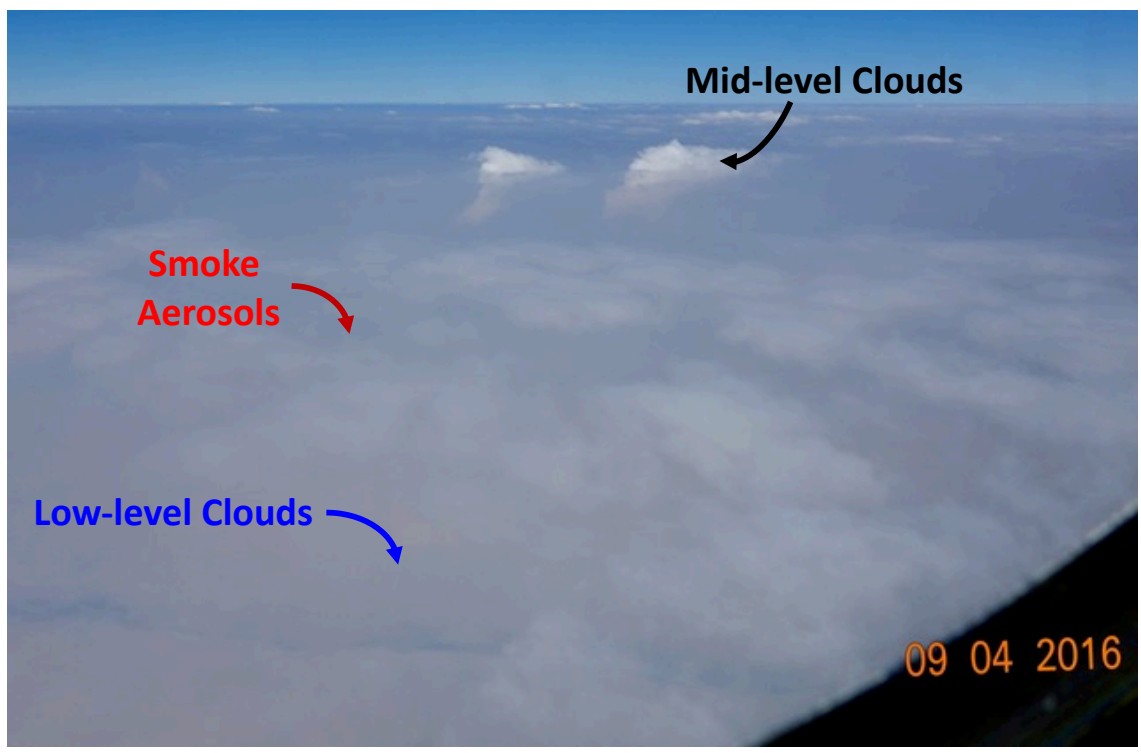

Figure 1: An Image taken during the NASA ORACLES Field campaign on September 4, 2016, showing mid-level clouds and smoke above the low-level clouds. Image taken by Paquita Zuidema.

710


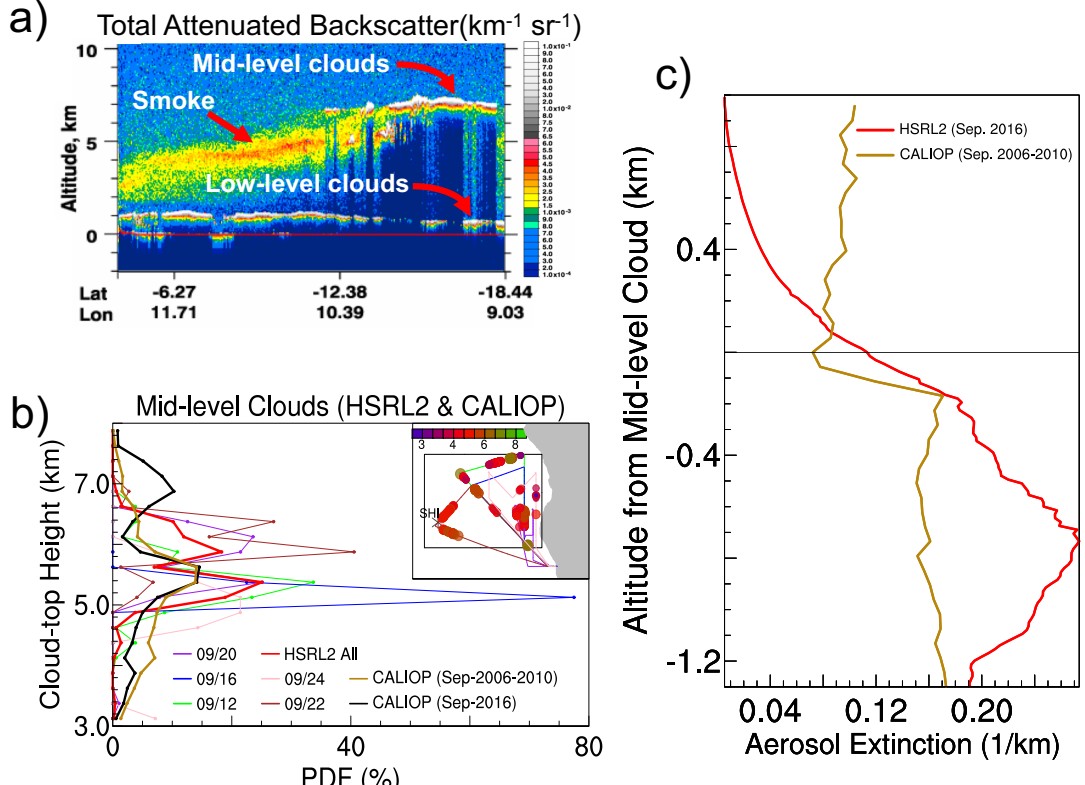

Figure 2: (a) An example image from CALIPSO showing CALIOP 532-nm total attenuated backscatter (km$^{-1}$ sr$^{-1}$)
with identifiable mid-level clouds, smoke, and low-level clouds on 22 September 2016 between ~00:54 UTC and
~00:57 UTC over the southeast Atlantic. (b) The probability distribution of mid-level cloud-top heights (km)
measured by the HSRL-2 on-board the ER-2 high-altitude aircraft during ORACLES in September 2016. The
combined distribution from HSRL-2 is shown by the thick red line, while the CALIOP distribution for all available
CALIPSO overpasses for September 2016 and September 2006-2010 are shown by the thick black and brown lines
respectively. The inset in (b) shows the spatial locations and heights (km) of the HSRL-2 mid-level cloud
measurements, as well as the region for the CALIOP distribution (5°-20°S and 10°W-12°E). (c) The 532-nm aerosol
extinction coefficients (km$^{-1}$) averaged for 0.2-degree grid box above and below the mid-level cloud top obtained
from HSRL-2 (red line; September 2016) and from CALIOP (brown line; September 2006-2010).





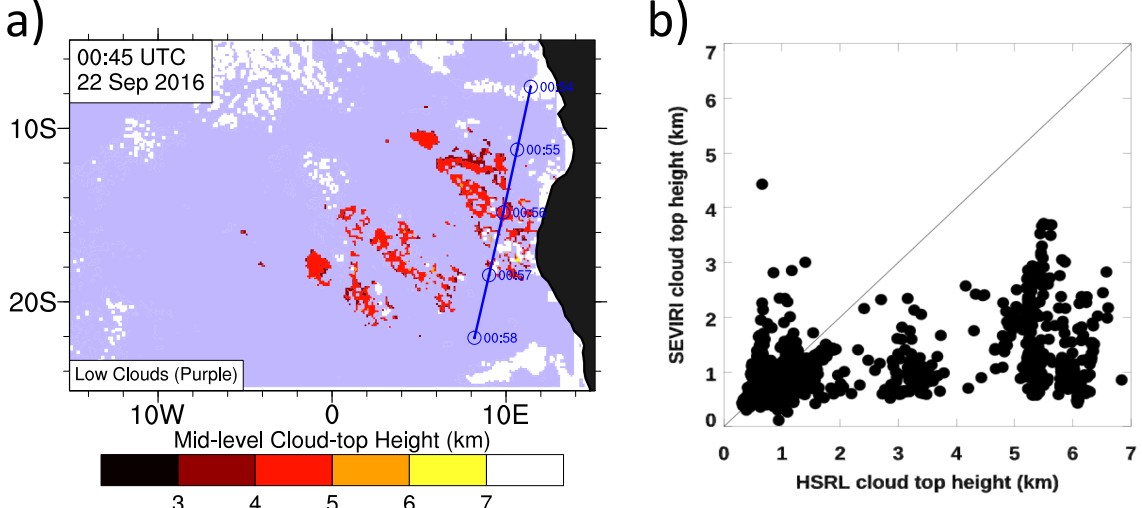

Figure 3: Comparison between mid-level clouds observed by SEVIRI and lidar-based instruments. (a) Image from SEVIRI instrument corresponding to the CALIPSO image in Figure 2a. This was taken 00:45 UTC, 22 September, 2016 and it shows the mid-level cloud-top heights (km, red-yellow shade), and the low-level clouds (purple; defined by cloud-top heights less than 3 km) over the southeast Atlantic. The blue line is the CALIPSO cross-over track for the image in Figure 2a, although it occurs 9 mins after the satellite image. (b) Comparison between SEVIRI and HSRL-2 cloud top height collocated within +/- 15 minutes of each other during ORACLES-2016.

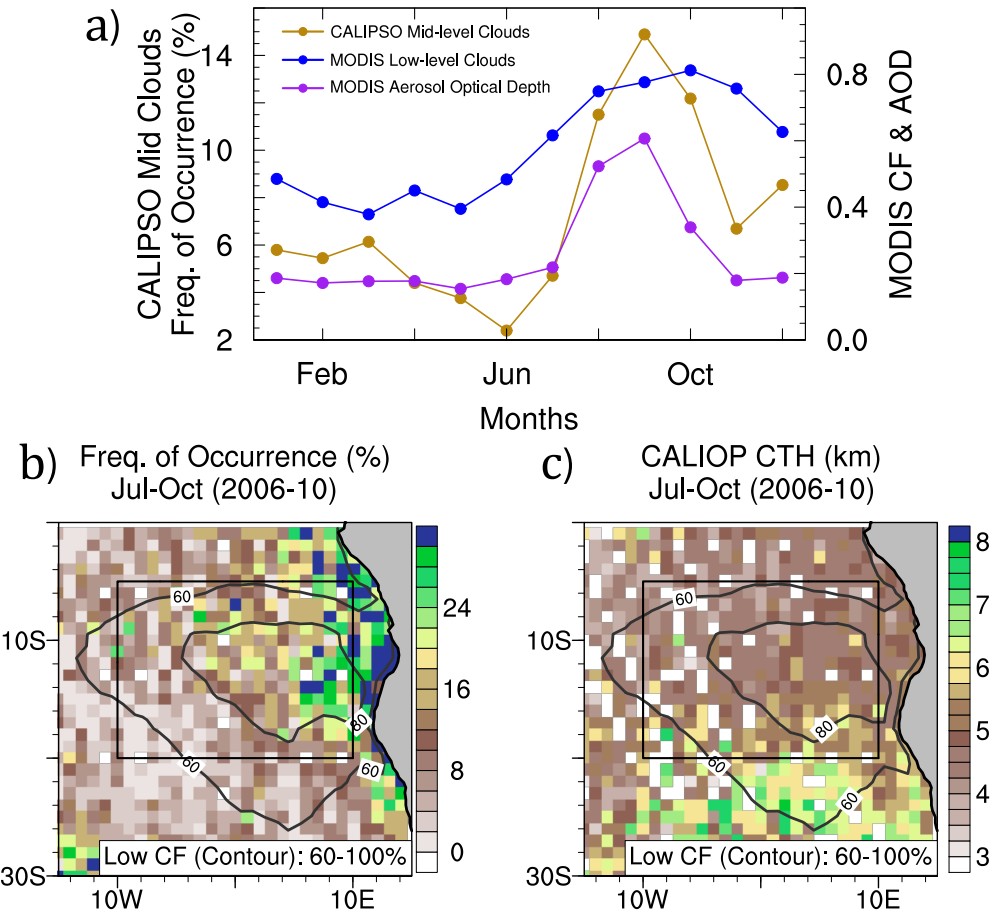


Figure 4: (a) Monthly averages (2006-2010) of the CALPSO mid-level cloud frequency of occurrence (%; brown line), MODIS low-level cloud fraction and aerosol optical depth (CF & AOD; right axis), all averaged over the ORACLES-2016 campaign region (defined here as 5º20ºS and 10ºW-10ºE; black boxes in Figure 3b & c). (b) The spatial distribution of the July–October average for the CALPSO mid-level cloud frequency of occurrence (%), and

(c) the corresponding cloud-top heights (km). The black contours in both Figure 3b & c are the MODIS liquid-water low-level cloud fraction (%) for the same period. The CALIPSO mid-level clouds are identified as cloud-layer top between 3–8 km, while the MODIS low-level clouds are averages of grid-boxes with cloud-top temperatures greater than 273 K.

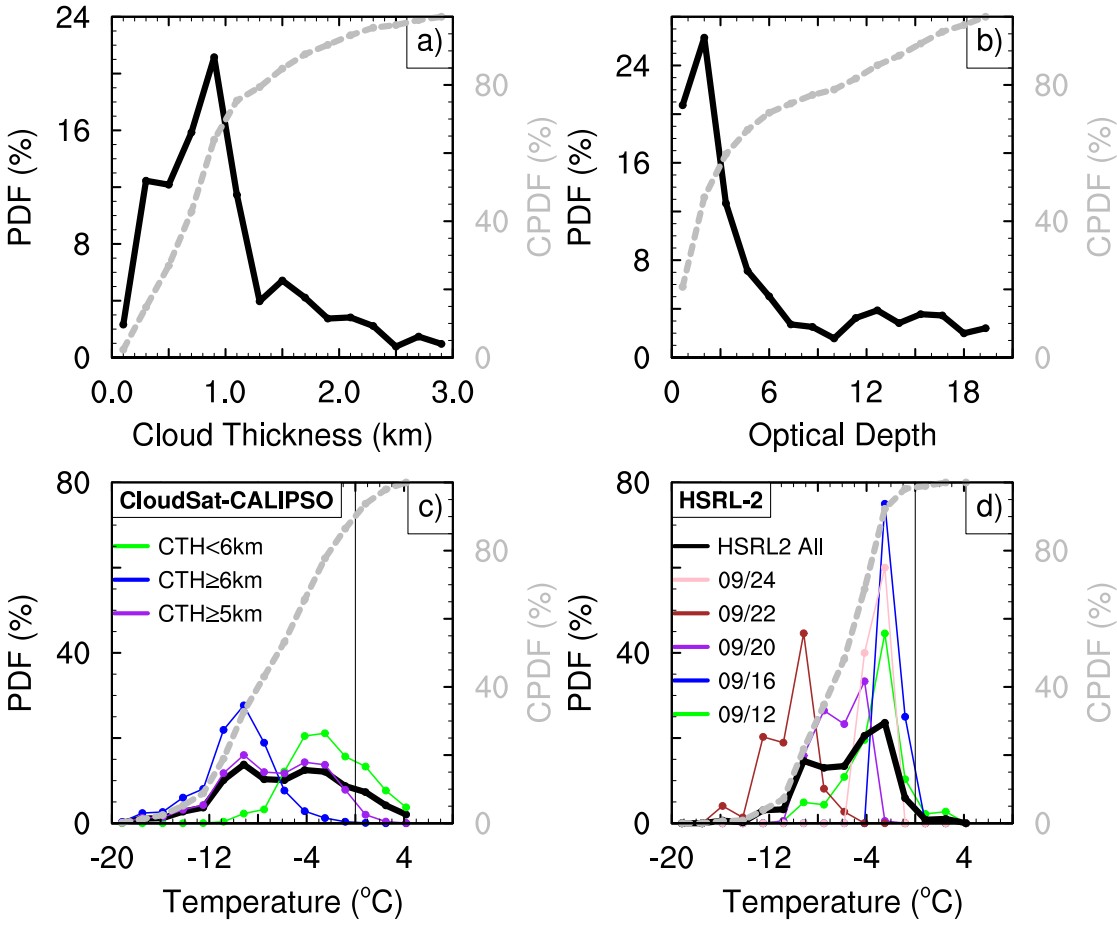

Figure 5: The probability (PDF; black solid lines) and cumulative (CPDF; grey dash lines) distributions of mid-level cloud properties. These distributions are obtained for mid-level (a) cloud thickness (km), (b) cloud optical depth, and (c) cloud temperature (ºC) from the CloudSat-CALIPSO merged dataset between 3–8 km altitude, July and October (2006-2010) averaged over the southeast Atlantic (black boxes shown in Figure 3). (c) also shows the temperature distribution subset into different cases of mid-level cloud-top heights. (d) Cloud temperature distribution (ºC) collocated with HSRL-2-derived mid-level clouds (see Figure 2b) and obtained for the individual days (colored lines) and the campaign period (HSRL2 All; black line for PDF and grey line for CPDF) during ORACLES in September, 2016. The thin vertical line in (c) and (d) shows the 0 ºC temperature.






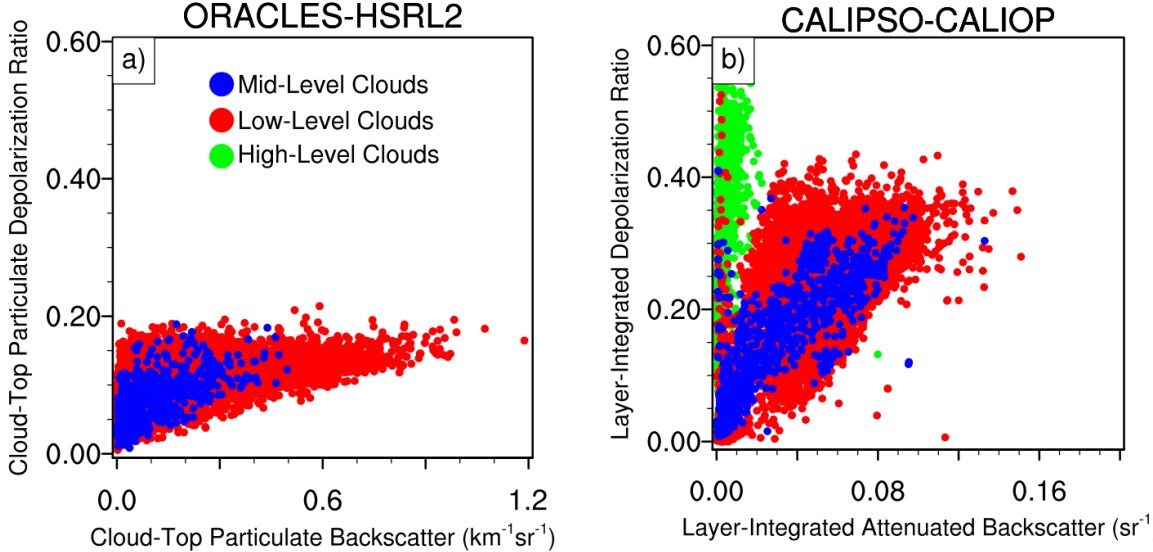

Figure 6: Identifying the phase of southeast Atlantic mid-level clouds. The left figure (a) shows the relationship between the particulate depolarization ratio and the particulate backscatter (km$^{-1}$sr$^{-1}$) at the cloud top obtained from HSRL-2 during ORACLES-2016 and right figure (b) shows the volume depolarization ratio and the attenuated
backscatter (sr$^{-1}$) integrated over the cloud layer during obtained CALIOP aboard CALIPSO. Both figures are estimated using available data during September 2016 and over the ORACLES-2016 campaign region (inset in Figure 2b). The low-level clouds (red dots) and high level clouds (green dots) respectively the observed clouds with cloud tops less than 3 km and greater than 8 km, while the mid-level clouds (blue dots) are the observed clouds with cloud tops between 3 and 8 km.


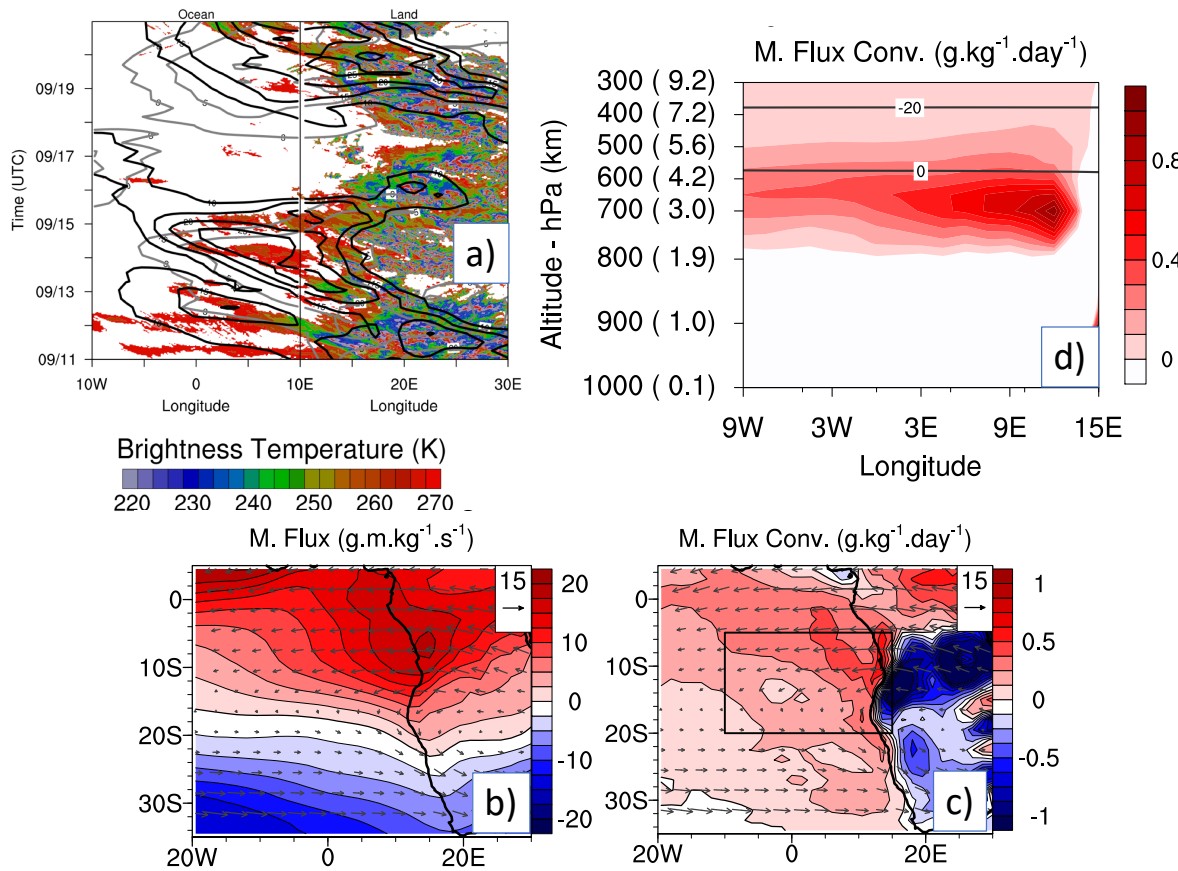

Figure 7: (a) An example showing the longitude-time cross-section of brightness temperature (K; shaded), easterly zonal wind speed (grey contours between 3-15 m s$^{-1}$ at 2 m s$^{-1}$ interval) and moisture flux (black contours between 10-30 g m kg$^{-1}$ s$^{-1}$ at 5 g m kg$^{-1}$ s$^{-1}$ interval) between 3–8 km and latitude range of 5°S-20°S for 11-20 September, 2016. The July–October (2006-2010) ERA-Interim (b) moisture flux (g.m.kg$^{-1}$.s$^{-1}$), and (c) moisture flux convergence (g.kg$^{-1}$.day$^{-1}$), averaged between 3–8 km. Positive is convergence and negative is divergence. The arrows are the moisture flux vectors, referenced at 15 g.m.kg$^{-1}$.s$^{-1}$. (d) The longitude-height transect of the moisture flux convergence, averaged between 5°S-20°S (black box in (b)). The horizontal lines in (d) represent the 0°C and -20°C isotherms averaged over the same period.






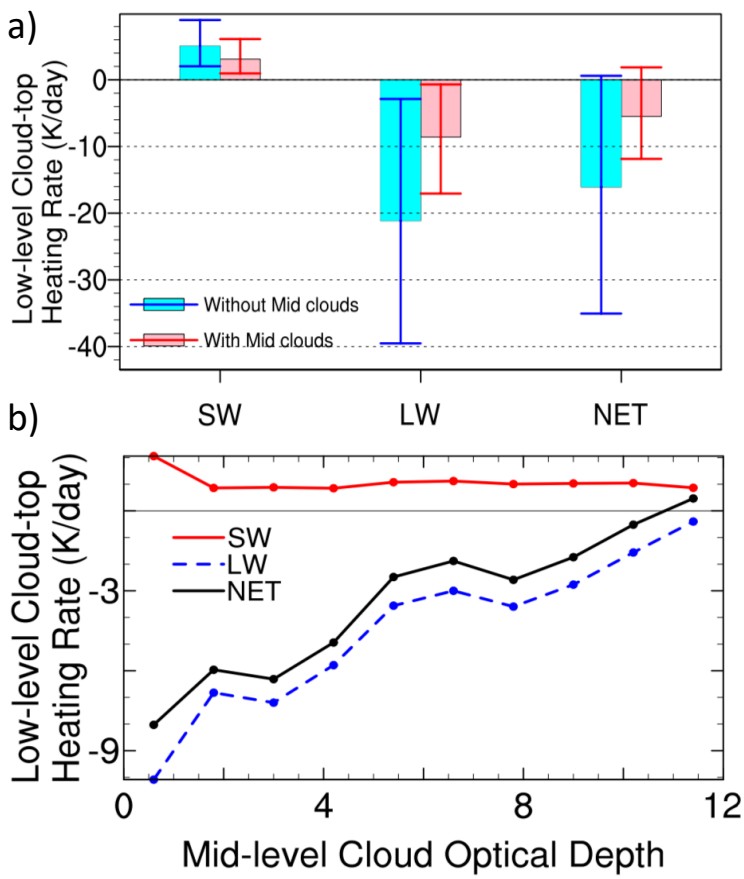

Figure 8: The radiative impact of mid-level cloud on the low-level cloud-top heating rates. (a) The instantaneous heating rates at the top of low-level clouds with (pink bars/red lines) and without (cyan bars/blue lines) the presence of collocated mid-level clouds. (b) The instantaneous heating rates at the top of the low-level clouds as a function of the overlying mid-level cloud optical depth. All data are obtained from the CALIPSO-CloudSat merged dataset between July and October (2006-2010), and over the ORACLES-2016 campaign region (black boxes shown in Figure 4) and separated into the shortwave (SW) and longwave (LW) components, as well as the NET (=SW+LW).







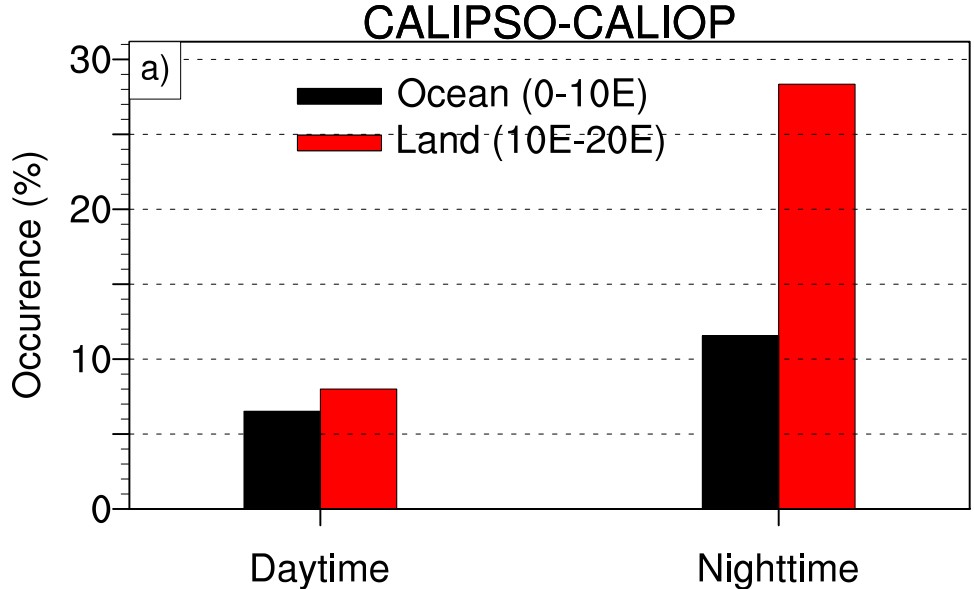

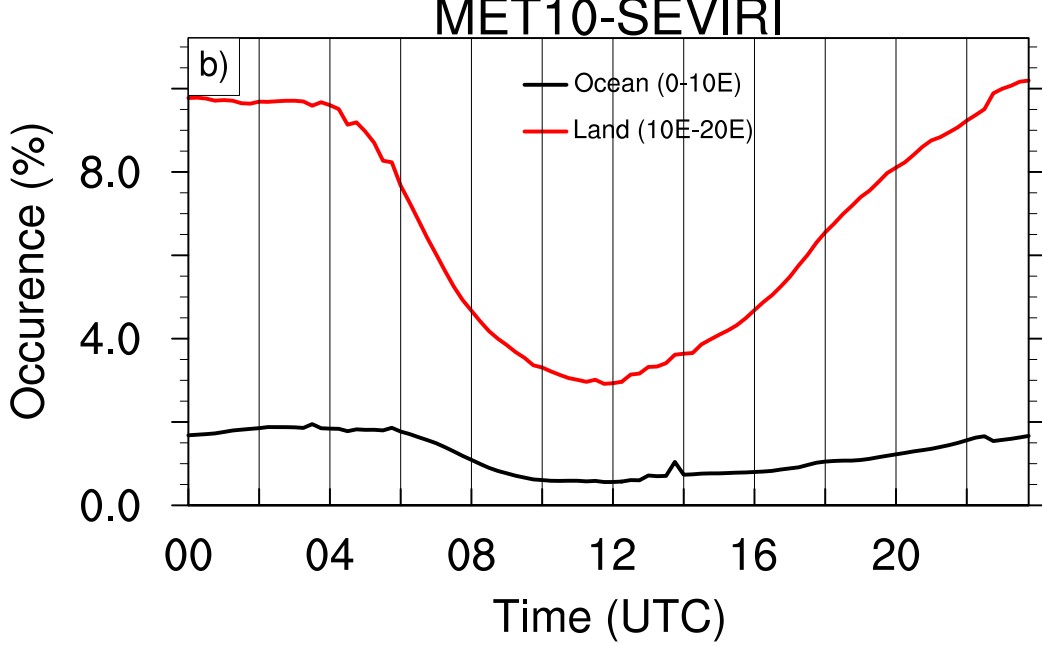

Figure 9: Diurnal variations of the mid-level cloud frequency of occurrence (%) between 3 and 8 km and 01-30 September, 2016 averaged for 5ºS-20ºS, over the ocean (black bar/line – 0-10ºE) and the land (red bar/line – 10ºE-20ºE) for observations obtained from (a) CALIOP instrument on board CALIPSO and (b) SEVIRI instruments onboard MET-10 satellite. CALIPSO overpass over the southeast Atlantic occurs between approximately 12:30-13:30 UTC during the day and 00:30-1:30 UTC during the night. Despite the difficulty of SEVIRI identifying the mid-level clouds, we use it here because of its higher temporal sampling (15 min) and only to give more insight into the structure of the diurnal variability and not necessarily its magnitude.