# Peer review of "Mid-level clouds are frequent above the southeast Atlantic stratocumulus clouds"

_Atmospheric Chemistry and Physics, 2020_

## Referee Comment (RC1) · Anonymous Referee #1 · 7 May 2020

Summary

This analysis uses several observational data sets, including aircraft observations from ORACLES, as well as CLAIPSO and CloudSat, to identify and quantify mid-level clouds off the coast of South Africa. Although low-level clouds (stratocumulus) in this region have received considerable attention, particularly in relation to smoke aerosols, mid-level clouds have not been properly investigated. The authors find that the occurrence of such mid-level clouds is relatively common, and go on to explore the meteorological causes, which includes synoptically-modulated continental moisture outflow. Finally, the authors quantify the corresponding radiative impacts of the south Atlantic mid-level

clouds.

Comments

Overall, the paper is very well written and the analysis is comprehensive and novel. My only comments pertain to model simulations, which were not part of this analysis. How well do models simulate these observed mid-level south Atlantic clouds? Can the results of this paper be used to understand possible model biases, and ultimately, improve model simulations in regard to this feature?

---

## Referee Comment (RC2) · Anonymous Referee #2 · 11 May 2020

The authors describe the occurrence, physical and optical properties, diurnal and seasonal variability, radiative impact on low-level clouds and the meteorology of mid-level clouds in the southeast Atlantic. For this purpose, they mostly used satellite observations and the in situ ORACLES 2016 campaign. The paper is clear and well written, and it makes good use of the observations. It is in particular of interest for the community that focuses on the southeast Atlantic region and it confirms the very few previous studies on the mid-level clouds. Therefore, I would recommend this paper for publication after a minor revision.

Minor comments:

[Figure]

Line 12: "no study" I find this statement a bit strong. It might be true that there are no studies about mid-level clouds in the southeast Atlantic region but there are mid-level cloud studies in the tropics (that you cited) that have similar findings as yours.

Line 13 & 423: "attribution" I do not find this word correct in both these sentences. I would rather write "estimation", "evaluation" or "calculation".

Line 13: "of the cloud radiation budget, as well as . . . retrievals" I would write first the retrieval and then the cloud radiation budget because first, the cloud properties are observed/retrieved and then, their radiation budget is evaluated.

Line 29: "to reducing" seems weird. Wouldn't be "to reduce" better?

Line 52-53: "Sub-tropical mid-level clouds have received little to no attention when. . ." To me, the subtropical region is the region between the tropics ($23.5°$) and the midlat-itudes. In this definition, southeast Atlantic as you defined it ($5°$S-$20°$S) is not in the subtropics. However, you might have a different definition. If so, please clarify because it is not clear to me. In addition, Riihimaki's paper focuses on Darwin ($12°$S) so it looks like it is the same latitude region as yours and Bourgeois's paper (2016) focuses on the tropics ($23.5°$N-$23.5°$S) so your region is encompassed in it. Therefore, I would not write that mid-level clouds described in your study have received no attention in the past but I would definitely agree with little attention. Same remark for the conclusion.

Line 101-102: "Despite the focus. . ." I do not really understand what the authors mean here. Do they speculate that their study applies to a larger region than the region they focus on? If so, why?

Line 107 & 431: Please write the altitude in international units first. Feet should not be used in research.

Line 138: Typo, "the the".

Line 165: CALIPO => CALIPSO.

[Figure]

Line 174: infra-red => infrared.

Line 184: It looks to me that "Therefore" is misused here.

Line 217: "southeast Atlantic".

Line 220: Write "5°S-20°S" for consistency.

Line 250: Typo, "Figure 5c & d".

Line 322: Typo, "for for".

Please, be consistent when you write longitudes, latitudes or temperatures. Use °
everywhere. Also, either you write e.g., 5km or 5 km. Several times, it is written "cloud"
instead of "clouds". References: Check them carefully. There are several mistakes.
Non-exhaustive examples: - Adebiyi A A A A A - Hobbs - Kacarab - Klein - Palikonda

Caption 4: CALPSO => CALIPSO (x2).

Caption 6: Verb missing in the last sentence.

---

## Author Comment (AC1) · 21 Jul 2020

We thank both reviewers for their comments, which have helped us to improve the paper. Below, we include a point-by-point response to the reviewer's comments and describe the corresponding changes we have made to the manuscript.

Reviewer: 1

This analysis uses several observational data sets, including aircraft observations from ORACLES, as well as CLAIPSO and CloudSat, to identify and quantify mid-level clouds off the coast of South Africa. Although low-level clouds (stratocumulus) in this region

have received considerable attention, particularly in relation to smoke aerosols, mid-level clouds have not been properly investigated. The authors find that the occurrence of such mid-level clouds is relatively common, and go on to explore the meteorological causes, which includes synoptically-modulated continental moisture outflow. Finally, the authors quantify the corresponding radiative impacts of the south Atlantic mid-level clouds.

Comments

Overall, the paper is very well written and the analysis is comprehensive and novel. My only comments pertain to model simulations, which were not part of this analysis. How well do models simulate these observed mid-level south Atlantic clouds? Can the results of this paper be used to understand possible model biases, and ultimately, improve model simulations in regard to this feature?

Reply: We thank the reviewer for the comment. Indeed, we did not emphasize the potential implication of our results to model simulations of mid-level clouds. For example, our result that showed the mid-level clouds are liquid-water clouds is useful to accurately simulate the phases of multi-layer clouds over the region. In addition, we also recognize that our results are useful for remote-sensing retrieval of both cloud and aerosols properties over the region. To highlight these points, we have added the paragraph to the Discussion & Conclusion section, which reads as follows:

"To that end, the prevalence of the multi-layer cloud system over the southeast Atlantic highlighted in this study could provide the needed guidance for future remote-sensing retrieval and any modeling efforts over the region. For example, the presence of the mid-level clouds must be accounted for in the observed top-of-the-atmosphere radiance received by the passive remote-sensing platforms (e.g., Peers et al., 2019), as well as the resulting retrieval of low-level cloud properties, including the low-level cloud-top heights over the region. In addition, the result that the southeast Atlantic mid-level clouds are supercooled liquid-water clouds could help reduce potential bias that may

[Figure]

be associated with the mixed-phase representation of mid-level clouds in the models (e.g., Zhang et al., 2005; Barrett et al., 2017). Overall, the knowledge of the mid-level cloud properties over southeast Atlantic could be useful to accurately simulate its radiative effects in the mid-troposphere, its impact on the underlying low-level clouds, the aerosol-cloud interaction, and, consequently, the regional cloud radiative budget."

Reviewer: 2

The authors describe the occurrence, physical and optical properties, diurnal and seasonal variability, radiative impact on low-level clouds and the meteorology of mid-level clouds in the southeast Atlantic. For this purpose, they mostly used satellite observations and the in situ ORACLES 2016 campaign. The paper is clear and well written, and it makes good use of the observations. It is in particular of interest for the community that focuses on the southeast Atlantic region and it confirms the very few previous studies on the mid-level clouds. Therefore, I would recommend this paper for publication after a minor revision.

Minor comments: Line 12: "no study" I find this statement a bit strong. It might be true that there are no studies about mid-level clouds in the southeast Atlantic region but there are mid-level cloud studies in the tropics (that you cited) that have similar findings as yours.

Reply: Thank you for the comments. We have re-phrased the sentence as follows:

"While much attention has focused on the interactions between the low-level clouds and the overlying aerosols, few studies have focused on the mid-level clouds that also occur over the region."

Line 13 & 423: "attribution" I do not find this word correct in both these sentences. I would rather write "estimation", "evaluation" or "calculation".

Reply: Thank you for the comment. We have replaced the word in both places with "evaluation".

Line 13: "of the cloud radiation budget, as well as . . . retrievals" I would write first the retrieval and then the cloud radiation budget because first, the cloud properties are observed/retrieved and then, their radiation budget is evaluated.

Reply: I have rephrased the sentence. The new sentence now reads:

"The presence of mid-level clouds over the region complicates space-based remote-sensing retrievals of cloud properties and the evaluation of the cloud radiation budget."

Line 29: "to reducing" seems weird. Wouldn't be "to reduce" better?

Reply: Yes. We have changed this part of the sentence. The sentence now reads:

"Clouds over the southeast Atlantic, as one of the world's major subtropical stratocumulus clouds (Klein and Hartmann, 1993), contribute importantly to the uncertainties in global climate change projections (Soden and Vecchi, 2011).

Line 52-53: "Sub-tropical mid-level clouds have received little to no attention when. . ." To me, the subtropical region is the region between the tropics (23.5âŮę) and the midlat- itudes. In this definition, southeast Atlantic as you defined it (5âŮęS-20âŮęS) is not in the subtropics. However, you might have a different definition. If so, please clarify because it is not clear to me. In addition, Riihimaki's paper focuses on Darwin (12âŮęS) so it looks like it is the same latitude region as yours and Bourgeois's paper (2016) focuses on the tropics (23.5âŮęN-23.5âŮęS) so your region is encompassed in it. Therefore, I would not write that mid-level clouds described in your study have received no attention in the past but I would definitely agree with little attention. Same remark for the conclusion.

Reply: Thank you for the comment. We recognize that our phrasing of the sentence may be too strong. We agree with the reviewer that other studies have broadly highlighted the mid-level clouds over latitudes that are comparable to that of the southeast Atlantic, although no major study (to our knowledge) focused specifically on this region.

To clarify the sentence, we have removed "to no", and it now reads: "Sub-tropical midlevel clouds have received less attention when. . .".

Line 101-102: "Despite the focus. . ." I do not really understand what the authors mean here. Do they speculate that their study applies to a larger region than the region they focus on? If so, why?

Reply: We thank the reviewer for the comment. Our study relies mainly on measurements that were taken during ORACLES campaign in 2016, which only covers a subset of the southeast Atlantic region. In other to give a broader context, we consider the mid-level clouds over the entire basin using satellite measurements. We have re-written the sentence to clarify this point as follows:

"Since this delimited area is part of the larger southeast Atlantic region, our analysis thus considers the entire southeast Atlantic region to provide a broader context for the occurrence of mid-level clouds beyond the area covered by ORACLES."

Line 107 & 431: Please write the altitude in international units first. Feet should not be used in research.

Thank you. We have changed this to kilometers in both places.

Line 138: Typo, "the the".

Thank you. This has been corrected.

Line 165: CALIPO => CALIPSO.

Thank you. This has been corrected.

Line 174: infra-red => infrared.

Thank you. All occurrence of this in the manuscript has been changed.

Line 184: It looks to me that "Therefore" is misused here.

We have removed it from the sentence.

Line 217: "southeast Atlantic".

Thank you. This has been corrected.

Line 220: Write "5âŮęS-20âŮęS" for consistency.

Thank you. This has been corrected.

Line 250: Typo, "Figure 5c & d".

Thank you. This has been corrected.

Line 322: Typo, "for for".

Thank you. This has been corrected.

Please, be consistent when you write longitudes, latitudes or temperatures. Use âŮę everywhere. Also, either you write e.g., 5km or 5 km. Several times, it is written "cloud" instead of "clouds". References: Check them carefully. There are several mistakes. Non-exhaustive examples: - Adebiyi A A A A A - Hobbs - Kacarab - Klein – Palikonda

We have looked through the references and made necessary corrections

Caption 4: CALPSO => CALIPSO (x2).

Thank you. This has been corrected.

Caption 6: Verb missing in the last sentence.

Thank you. This has been corrected.

References: Barrett, A. I., Hogan, R. J. and Forbes, R. M.: Why are mixed-phase altocumulus clouds poorly predicted by large-scale models? Part 1. Physical processes, J. Geophys. Res. Atmos., 122(18), 9903–9926, doi:10.1002/2016JD026321, 2017. Peers, F., Francis, P., Fox, C., Abel, S. J., Szpek, K., Cotterell, M. I., Davies, N. W., Langridge, J. M., Meyer, K. G., Platnick, S. E. and Haywood, J. M.: Observation of absorbing aerosols above clouds over the south-east Atlantic Ocean from the

**ACPD**

geostationary satellite SEVIRI – Part 1: Method description and sensitivity, Atmos. Chem. Phys., 19(14), 9595–9611, doi:10.5194/acp-19-9595-2019, 2019. Zhang, M. H., Lin, W. Y., Klein, S. A., Bacmeister, J. T., Bony, S., Cederwall, R. T., Del Genio, a. D., Hack, J. J., Loeb, N. G., Lohmann, U., Minnis, P., Musat, I., Pincus, R., Stier, P., Suarez, M. J., Webb, M. J., Wu, J. B., Xie, S. C., Yao, M. S. and Zhang, J. H.: Comparing clouds and their seasonal variations in 10 atmospheric general circulation models with satellite measurements, J. Geophys. Res. D Atmos., 110(15), 1–18, doi:10.1029/2004JD005021, 2005.

Please also note the supplement to this comment:
https://www.atmos-chem-phys-discuss.net/acp-2020-324/acp-2020-324-AC1-supplement.pdf